# Effect of Male Body Size on Female Reproduction in *Pyrrhocoris apterus* (L.) (Heteroptera, Pyrrhocoridae)

**DOI:** 10.3390/insects13100902

**Published:** 2022-10-03

**Authors:** Alois Honek, Zdenka Martinkova

**Affiliations:** Crop Research Institute, Functional Diversity in Agro-Ecosystems, Drnovská 507, Ruzyně, 161 06 Praha 6, Czech Republic

**Keywords:** assortative mating, oviposition, fecundity, egg number, egg size

## Abstract

**Simple Summary:**

The benefits of copulation with large males were investigated in the firebug, *Pyrrhocoris apterus* (L.) (Heteroptera). Sexually immature adults were transferred from the field and kept as female-male pairs with different partner size ratios. We investigated the influence of male body size on the number and weight of eggs and time elapsed from the transfer to their deposition. The expected positive effect of increasing male body size on the characteristics of female reproduction was not detected.

**Abstract:**

Females and males of the abundant heteropteran species *Pyrrhocoris apterus* (L.) mate with the larger of the available partners. The male benefits from copulation with a large female, which is more productive than a small female. However, the benefit to females from copulation with a large male has not yet been investigated. Overwintered sexually immature adults were transferred from the field to indoors for a long day at 25 °C and subsequently kept as female-male pairs with different partner size ratios. The female lays eggs in several successive clutches. We investigated the influence of male size on the number and weight of eggs in individual clutches and the time elapsed from the transfer to their deposition. These characteristics of reproduction were first regressed on female size, and residuals of this regression were then regressed on male size. The positive effect of increasing male size on the characteristics of female reproduction manifested as a significant value of the latter regressions. The expected positive effect of increasing male body size on female reproductive characteristics was not detected. Several reasons for this deficiency are discussed.

## 1. Introduction

Body size significantly affects several characteristics and life functions of animals at the interspecific and intraspecific levels [1,2,3]. In insects, intraspecific variability of body size can affect lifespan [4,5], resistance to adverse external conditions [6,7], movement abilities [8,9] and several other characteristics. Therefore, body size also affects reproductive abilities, particularly those of females—for example, the number of ovarioles [10], number of oocytes [11] or embryos [12] in the ovariole, the frequency and degree of autogeny [13], size of eggs [14] or rate of their deposition [15].

Because body size significantly affects reproductive functions, it also plays an essential role in the selection of sexual partners. Males select the larger available females [16]. Their benefit resulting from obtaining a large female is an increased quantity and quality of offspring because female fecundity increases with body size [17] and the offspring of a large female are often of higher quality than those of a small female [18]. The female benefits from mating with the larger of the offered males because the larger male guarantees a larger quantity and, in some cases, better quality of sperm than a small male [19,20,21]. Thus, mating with larger males ensures the fertilization of a larger number of offspring and/or reduces remating frequency [22]. However, the advantage resulting from copulation with a larger partner can also affect the female. The female receives a larger quantity of liquid secreted by the accessory glands from a large male than it would receive from a small male [23], and this support from the male may contribute to her own benefit, i.e., increasing female reproductive performance (e.g., egg number or size). Research aimed at identifying these effects is still scarce [24,25].

The role of female preference in selecting a male partner is unclear because female preference can only partially influence the choice of mate, as the configuration of the pair is influenced by the courtship and mating behaviour of the male. Large males have an advantage over small males in male-male conflicts [26,27,28,29] or they are more successful in coercing the female to mate [30,31,32]. Therefore, copulation with a large male is often not the result of female preference but the activity and preference of the male [33,34]. However, in many cases, the female truly prefers large males, for example, because males manifest their superior fitness through courtship displays or nuptial gifts [35,36,37,38]. The superiority of large males and the answer to why they are preferred by females may remain hidden because they have not yet been experimentally determined [39,40,41]. Thus, the adaptive significance of male choice for large females is a direct reproductive gain, while the advantage to females of choosing large males is less clear [42].

The effect of male size on the viability and fecundity of females has been little studied thus far. A suitable subject for these studies is *Pyrrhocoris apterus* (L.) (Heteroptera, Pyrrhocoridae). This species is a pre- and post-dispersal seed predator, feeding mainly on Malvaceae and *Tilia* spp. seeds [43]. Large local populations of *P. apterus* live around the host plants. In Central Europe, sexually immature adults overwinter. After hibernation, mating occurs, and the females deposit eggs in clutches that are laid at several-day intervals. Body size may play a role in the mutual selection of mating partners. In laboratory experiments, females were observed to prefer larger offered males [44], but the benefit they derive from copulation with a large male has not been investigated. The size of accessory glands in males of *P. apterus* is large, with a length of approximately 2 mm [45], which is more than one-fifth of the average length of the male’s body. The size of the accessory glands is generally positively correlated with the size of the male [46,47,48]. We hypothesized that the size of the accessory glands increasing in parallel with the size of the male will positively affect the vital functions and reproductive characteristics of the female. This positive effect was observed in *P. apterus* in cases where the size of the accessory glands changed with the age of the male or depending on the wing form of the male [49]. Based on this assumption, two experiments were conducted (in 2004 and 2022), the aims of which were to determine the influence of the size of the male partner on the reproductive functions of the female.

## 2. Materials and Methods

Adults for the experiments were collected at two localities in the Czech Republic; in the 2004 experiment, on March 9 at the locality of Lounín (49.9076 N, 14.0114 E, 405 m a.s.l.); in the 2022 experiment, on February 5 at the locality of Stará Lysá (50.2218 N, 14.8037 E, 189 m a.s.l.). In both cases, they originated from natural populations hidden during overwintering in the litter near the feet of linden trees (*Tilia cordata* L.). The experimental animals were collected before the start of the period of spring mating. From collection until the beginning of the experiment, these adults were kept at 5 °C in the dark.

On the day the experiments were established, 11 March 2004 and 16 February 2022, the body length of the experimental individuals was determined by measuring from the tip of the rostrum to the hind margin of the last abdominal tergite with an accuracy of 0.5 mm. In both terms, 100 males with a body length of 8.0–11.0 mm and 100 females with a body length of 8.5–11.5 mm were included in the experiment, in which 100 female-male pairs were created each year. We supposed that the influence of male size on female oviposition would be more pronounced when the difference in size between the two partners in the pair was greater. Therefore, small males were preferentially paired with large females and large males with small females, and additionally males with females of similar size. Of the 100 originally established pairs, 57 survived to the beginning of oviposition in 2004, and 45 survived to the beginning of oviposition in 2022 (Table 1). Pairs established at the beginning of the experiment were kept together for the entire duration of the experiment, until the death of one of the partners.

In the laboratory experiments, adults were kept in a climatized room with a constant 25 °C temperature and a 16 h light / 8 h dark photoperiod. Each pair was placed in a cylindrical transparent plastic cup (height, 8 cm; diameter, 5.5 cm) closed on top by nylon fabric. Food (*T. cordata* seeds) were scattered in excess at the bottom of the cup and replaced at weekly intervals. Water was provided on a wet piece of cotton wool. For each female, each egg clutch, the order of the clutch, number of eggs laid in the clutch, the weight of the eggs and term of its deposition were recorded (i.e., the time that elapsed from the date of establishment of the experiment to the date of laying of the clutch). The presence of newly laid clutches was recorded at ~12 h intervals, and inspections of rearing cups occurred at ~07:00 and ~19:00 Central European Time. Egg clutches were weighed within 24 h after deposition, with an accuracy of 10^−4^ g.

The following oviposition characteristics were monitored for each female: (1) number of eggs in the clutches of the particular order from the beginning of oviposition, (2) cumulative number of eggs laid from the beginning of oviposition up to the deposition of the clutch of the particular order, (3) average weight of one egg in the clutch of the particular order, and (4) elapsed time from the date of establishment of the experiment to the date of laying of the clutch. Characteristics (1) and (2) were recorded in both experiments of 2004 and 2022, and characteristics (3) and (4) were recorded only in the experiment of 2022. We present the data for the first five clutches when the number of experimental pairs was >20.

All the observed characteristics of oviposition (1)–(4) can be influenced both by female and male size. This study aims to determine the influence of male size on particular characteristics of female reproduction. To achieve this aim, a standard procedure was used: (a) for a given characteristic of female reproduction, the regression of the set of values of this characteristic on the body length (in mm) of the female (Lf) was calculated, and the residual values of this regression were calculated. Next, we calculated (b) the regression of these residuals on the body length (in mm) of the male (called absolute size, La) and (c) regression of these residuals on the relative size of the male (Lr), which is the difference between the male and female body length calculated as Lr = La − Lf. We assumed that finding a significant relationship in the latter regressions (b) and (c) would indicate the influence of male size on the given characteristic of female reproduction. As an example for visual demonstration of this procedure the results for the first clutch in the 2004 experiment (shown in Figure 1) were chosen. All the calculations were performed using SigmaStat 3.5 [50].

## 3. Results

Effect of male size on the number of eggs: in the 2004 and 2022 experiments, a significant positive relationship was found between female size and number of eggs laid (except clutch 5). However, no significant relationship was found between the residuals of this regression and absolute (La) or relative (Lr) size of the male, either in sets of clutches of a particular order or number of eggs aggregated since the start of oviposition (Figure 1, Table 2).

Effect of male size on the weight of eggs: In the 2022 experiment, neither a significant regression of egg mass on female size (except results for the first clutch) nor a significant relationship between the residuals of this regression and absolute (La) or relative (Lr) size of the male was established (Table 3).

Effect of male size on the oviposition rate: in the 2022 experiment, neither a significant regression of the time elapsed from the beginning of the experiment (transfer to laboratory conditions) to laying of the clutch on the size of the female nor a significant relationship between the residuals of this regression and absolute (La) or relative (Lr) size of the male was established (Table 4).

## 4. Discussion

To assess the study results, considering first the specific conditions of courtship and mating in natural populations of *P. apterus* is useful. The copulation behaviour of this species has often been studied experimentally in the laboratory [44]. When two males of contrast size were offered to a female, the larger of two males was more likely to mate in cases when the experimental female was large, but not when the experimental female was small. In addition, when two females and a male were used, the larger female was the one more likely to mate, but only if the male was large. This suggests that both large males and large females have a preference (or other mating advantage) with each other, but smaller individuals have little preference, or even a preference for smaller mates. furthermore, other studies of copulatory behaviour have been made in the laboratory [45,51,52,53] or under seminatural conditions [54] but not in the wild. The reason is that local populations, often comprising thousands of individuals, are aggregated in small areas delimited by a suitable microclimate. The adults aggregate there in early spring, simultaneously at the beginning and peak of their sexual activity [55]. In these aggregations, successful and unsuccessful mating attempts of males, as well as the reactions of females, which range from passive acceptance of the male to a rejection that comprises avoiding contact or a short escape, can be observed. The description of individual types of male/female interactions, frequency of different modes of mating behaviour and body size of the involved females and males has not yet been performed. As in other pyrrhocorids [31], the outcome of these contacts is likely influenced by the partner size. To what extent the choice of mate is determined by the different intensities of aggressive and/or coercive behaviour of large and small males and to what extent the outcome depends on the receptive or repulsive response of females remain unclear. Considering the influence of male body size on female reproduction would be interesting if the choice of partner was influenced by the female. Because these questions cannot be answered based on observations in nature, the effect of male size on female reproduction must be investigated experimentally.

In our experiments, no significant relationship was found between male size and any of the investigated reproductive characteristics of the female, the number of eggs, the weight of the eggs or the rate of their laying. The lack of an effect of male size on female reproduction can be explained in different ways. First, the choice of the male is not influenced by female preferences. Large males may have a greater ability to coerce mating with a female than small males, and mating with a large male is then a result of its superior ability to assert itself without benefiting the female. Indeed, under experimental conditions, large males were recorded more often in copulation than small males when the female was large but not when the female was small [44], a finding that can be interpreted as large males being more active and successful in male-male competition if they are rewarded by copulation with a large female.

Furthermore, large or small body size is not a sufficient “marker” of male quality in natural populations. The reason may be that, in a natural population, environmental conditions act against the development of extremely large and small individuals. Under laboratory conditions of 25 °C and food ad libitum, wild strain males grow to an average size of 0.5 to 2.0 mm larger than those in natural populations [56]. By contrast, starvation and low temperature during the final period of larval development substantially reduce the body size of males [57,58]. However, these dwarf individuals die during the winter period [54]. A significant effect of male body size on female reproduction would be expected if extremely large and small males were used for the experiments. However, these extremely large or small males were unavailable in natural populations.

Finally, we speculate that males bring different advantages for the reproduction of females than those investigated in our experiment. In females isolated from males, the sperm supply is sufficient to fertilize up to seven batches, but the percentage of fertilized eggs gradually decreases with the order of the clutch [54]. The advantage of mating with a large male can be a large supply of sperm that can be used during periods when the possibility of copulation is limited or excluded. However, in natural populations, repeated mating often occurs, due to the large size of local populations and the high frequency of copulation.

## 5. Conclusions

In *P. apterus* individuals from natural populations, the influence of male size on any of the reproductive characteristics of the female (the number of eggs, the weight of the eggs or the rate of their laying) was not revealed. However, the influence of male body size on female reproduction is possible. Its effect could manifest if males of extreme sizes were available in the population. Alternatively, its effect could manifest if different characteristics of female reproduction than those studied in our experiments were studied. Thus, more reasons may explain why these effects were not detected in our experiments. The topic should be the subject of further research.

## Figures and Tables

**Figure 1 insects-13-00902-f001:**
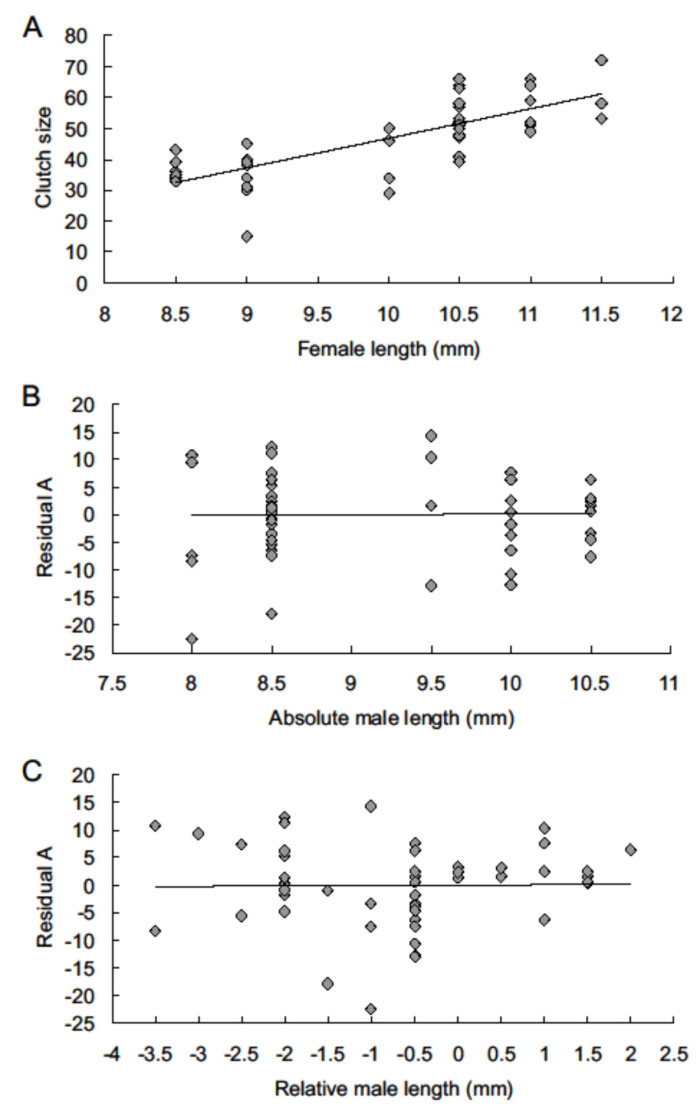
The relationship between fecundity and body size in the first clutch laid in 2004 experiment. (**A**): Regression of the number of eggs on female body length (y = 9.55x − 48.52, R^2^ = 0.5969). (**B**): Regression of the residuals of regression A on absolute male size La (male body length, mm) (y = 0.17x − 1.59, R^2^ = 0.0004). (**C**): Regression of the residuals of regression A on relative male size Lr (Lr = male body length – female body length) (y = 0.08x + 0.06, R^2^ = 0.0002).

**Table 1 insects-13-00902-t001:** Size of males and females put together in experimental pairs. The figures indicate the number of pairs that survived until the start of the oviposition, in the 2004 and 2022 experiments.

		2004							
		Male Size (mm)							
		8	8.5	9	9.5	10	10.5	11	Sum
Female size (mm)	8.5		3		1	1	1		6
	9	2	8		1	4	3		18
	9.5								0
	10		2		1		1		4
	10.5		12		1	7			20
	11	1	2				3		6
	11.5	2					1		3
	Sum	5	27	0	4	12	9	0	57
		**2022**							
		**Male Size (mm)**							
		8	8.5	9	9.5	10	10.5	11	Sum
Female size (mm)	8.5	2	1			2		1	6
	9					9			9
	9.5		1	1	3		3		8
	10			1	2				3
	10.5		6	1			4	1	12
	11	1	2		1	1			5
	11.5		1		1				2
	Sum	3	11	3	7	12	7	2	45

**Table 2 insects-13-00902-t002:** Relationship between body size and number of eggs deposited in the first five clutches, in the experiments of 2004 and 2022. Clutch: clutch order - the clutches of the particular order and aggregated numbers of eggs laid from the beginning of oviposition up to the deposition of the clutch of the particular order (e.g., 1+2 indicates aggregated number of eggs in clutches 1 and 2, etc.), N: the number of females that laid clutches of the particular order. Regression: (a) Female size: regression of number of eggs on female body length (mm). (b) Male size (abs): regression of the residuals of regression (a) on “absolute male size”, which is male body length (mm). (c) Male size (rel): regression of the residuals of regression (a) on “relative male size”, which is the difference between male and female body length (mm). a: slope of the regression line, Rsqr: coefficient of determination of the regression line, F: F statistic of the ANOVA test, P: P value of the ANOVA test.

				Regression											
Clutch		n Eggs		(a) Female Size				(b) Male Size (abs)				(c) Male Size (rel)			
	N	Mean ± SE		a	Rsqr	F	P	a	Rsqr	F	P	a	Rsqr	F	P
2004															
1	57	45.9 ± 1.52		9.549	0.5970	81.44	<0.001	0.174	0.0004	0.02	0.877	0.079	0.0002	0.01	0.917
2	57	53.1 ± 1.38		7.717	0.4750	49.84	<0.001	−0.222	0.0007	0.04	0.848	−0.101	0.0003	0.02	0.897
3	57	46.8 ± 1.63		6.522	0.2420	17.56	<0.001	0.799	0.0043	0.24	0.628	0.364	0.0020	0.11	0.743
4	42	37.8 ± 2.24		7.308	0.2110	10.69	0.002	2.273	0.0238	0.98	0.329	1.017	0.0107	0.43	0.515
5	21	31.2 ± 2.69		3.072	0.0479	0.96	0.341	1.049	0.0062	0.12	0.735	0.531	0.0031	0.06	0.810

1 + 2	57	98.9 ± 2.70		17.266	0.6220	90.43	<0.001	−0.048	0.0000	0.00	0.980	−0.022	0.0000	0.00	0.987
1 + 2 + 3	57	145.7 ± 3.90		23.788	0.5650	71.54	<0.001	0.751	0.0012	0.06	0.801	0.343	0.0005	0.03	0.865
1 + 2 + 3 + 4	42	184.9 ± 5.37		30.557	0.5770	54.56	<0.001	3.730	0.0187	0.76	0.387	1.668	0.0084	0.34	0.564
1 + 2 + 3 + 4 + 5	21	217.5 ± 8.96		32.143	0.4740	17.11	<0.001	7.027	0.0454	0.90	0.354	3.555	0.0230	0.45	0.512
2022															
1	45	44.1 ± 1.30		4.269	0.1800	10.32	0.002	1.761	0.0339	1.65	0.205	0.648	0.0125	0.60	0.444
2	44	48.8 ± 1.73		5.005	0.1420	6.98	0.012	1.963	0.0255	1.10	0.301	1.126	0.6960	0.38	0.540
3	43	52.3 ± 1.60		2.683	0.0489	2.11	0.154	3.381	0.0830	3.71	0.061	1.222	0.0300	1.27	0.267
4	40	54.7 ± 1.50		4.916	0.2050	9.79	0.003	−1.848	0.0340	1.34	0.255	−0.662	0.0122	0.47	0.498
5	30	51.5 ± 2.18		6.098	0.1680	6.88	0.013	−0.730	0.0025	0.09	0.771	−0.229	0.0008	0.03	0.871

1 + 2	44	93.6 ± 2.62		9.189	0.2090	11.11	0.002	4.195	0.0549	2.44	0.126	1.490	0.0195	0.84	0.366
1 + 2 + 3	43	147.1 ± 3.65		10.56	0.1450	6.96	0.012	7.830	0.0949	4.30	0.044	2.829	0.0343	1.46	0.234
1 + 2 + 3 + 4	40	204.0 ± 4.57		15.836	0.2060	9.59	0.004	3.900	0.0145	0.55	0.465	1.280	0.0048	0.18	0.676
1 + 2 + 3 + 4 + 5	30	251.6 ± 6.24		21.15	0.2580	11.46	0.002	2.090	0.0028	0.09	0.762	0.644	0.0009	0.03	0.866

**Table 3 insects-13-00902-t003:** Relationship between body size and egg mass in the first five clutches laid after the start of the oviposition, experiment of 2022. Clutch: clutch order, N: number of clutches. Egg mass: weight of one egg (g*10,000). Regression: (a) Female size: regression of egg mass on female body length (mm). (b) Male size (abs): regression of residuals of regression (a) on “absolute male size” which is male body length (mm). (c) Male size (rel): regression of residuals of regression (a) on “relative male size” which is the difference between the male and female body length (mm). a: slope of regression line, Rsqr: coefficient of determination of regression line, F: F statistics of ANOVA test, P: P value of ANOVA test.

			Regression											
Clutch		Egg Mass	(a) Female Size				(b) Male Size (abs)				(c) Male Size (rel)			
	N	mean ± SE	a	Rsqr	F	P	a	Rsqr	F	P	a	Rsqr	F	P
1	45	3.80 ± 0.070	0.296	0.3180	20.02	<0.001	0.037	0.0068	0.30	0.590	0.014	0.0026	0.11	0.738
2	44	3.61 ± 0.059	0.074	0.0269	1.13	0.293	−0.112	0.0615	2.69	0.109	−0.040	0.0218	0.91	0.345
3	43	3.61 ± 0.052	0.053	0.0193	0.71	0.405	−0.070	0.0383	1.44	0.239	−0.025	0.0129	0.49	0.491
4	40	3.55 ± 0.048	0.081	0.0659	2.40	0.131	0.037	0.0120	0.41	0.525	0.012	0.0039	0.13	0.718
5	30	3.38 ± 0.082	0.021	0.0017	0.05	0.825	0.093	0.0251	0.75	0.395	0.028	0.0075	0.22	0.644

**Table 4 insects-13-00902-t004:** Relationship between body size and the time elapsed to deposition of the first five clutches, experiment 2022. Clutch: clutch order, N: number of clutches. To (days): the time elapsed from the transfer to laboratory conditions to deposition of the clutch (days). Regression: (a): Female size: regression of To on female body length (mm). (b) Male size (abs): regression of residuals of regression (a) on “absolute male size” which is male body length (mm). (c) Male size (rel): regression of residuals of regression (a) on “relative male size” which is the difference between the male and female body length (mm). a: slope of regression line, Rsqr: coefficient of determination of regression line, F: F statistics of ANOVA test, P: P value of ANOVA test.

			Regression											
Clutch		To (Days)	(a) Female Size				(b) Male Size (abs)				(c) Male Size (rel)			
	N	mean ± SE	a	Rsqr	F	P	a	Rsqr	F	P	a	Rsqr	F	P
1	45	7.4 ± 0.15	0.188	0.0273	1.29	0.262	0.053	0.0021	0.10	0.758	0.020	0.0008	0.04	0.850
2	44	12.2 ± 0.23	0.406	0.0542	2.35	0.133	0.088	0.0027	0.11	0.739	0.032	0.0010	0.04	0.841
3	43	18.2 ± 0.41	0.445	0.0205	0.86	0.360	0.295	0.0094	0.39	0.537	0.107	0.0034	0.14	0.711
4	40	23.2 ± 0.59	0.893	0.0423	1.72	0.197	0.735	0.0287	1.15	0.290	0.265	0.0103	0.41	0.527
5	30	28.0 ± 0.77	0.257	0.0025	0.08	0.776	1.113	0.0390	1.34	0.256	0.343	0.0120	0.40	0.531

## Data Availability

The data presented in this study are available on request from the corresponding author.

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
