# Peer review of "Effect of Male Body Size on Female Reproduction in Pyrrhocoris apterus (L.) (Heteroptera, Pyrrhocoridae)"

_insects, 2022, doi:10.3390/insects13100902_

Round 1

Reviewer 1 Report

The authors of this manuscript examined the effect of male body size on the reproductive output of their partner females in a bug. No significant effect was observed. This is a negative report for the hypothesis that the male size produces benefits. However, such a negative result should be published somewhere. Therefore, I recommend the manuscript for publication in Insects. I have two comments:

 (1) If the authors consider the possibility of female choice, the performance of the progeny should also be examined. Females choose large males and the body size reflect their genetic quality, their progeny show higher performance such as nymphal growth rare and reproductive output.

 (2) The authors used various sizes of females also. I understand that the statistical analysis can pick out male’s effect. However the female size may possibly make the results obscure. If they use a stable size of females, the results might be clearer.

Author Response

Reviewer 1

English language and style

( ) Extensive editing of English language and style required
( ) Moderate English changes required
(x) English language and style are fine/minor spell check required
( ) I don't feel qualified to judge about the English language and style

Yes

Can be improved

Must be improved

Not applicable

Does the introduction provide sufficient background and include all relevant references?

(x)

( )

( )

( )

Are all the cited references relevant to the research?

(x)

( )

( )

( )

Is the research design appropriate?

(x)

( )

( )

( )

Are the methods adequately described?

( )

(x)

( )

( )

Are the results clearly presented?

(x)

( )

( )

( )

Are the conclusions supported by the results?

(x)

( )

( )

( )

Comments and Suggestions for Authors

Minor changes to English have been corrected according to suggestions of  reviewers. The English was edited before sending to Insects, with the help of American Journal Experts. We attach a certificate of editing.

The authors of this manuscript examined the effect of male body size on the reproductive output of their partner females in a bug. No significant effect was observed. This is a negative report for the hypothesis that the male size produces benefits. However, such a negative result should be published somewhere. Therefore, I recommend the manuscript for publication in Insects. I have two comments:

 (1) If the authors consider the possibility of female choice, the performance of the progeny should also be examined. Females choose large males and the body size reflect their genetic quality, their progeny show higher performance such as nymphal growth rare and reproductive output.

Many thanks for a valuable comment. Offspring performance was not studied, but we will certainly consider this issue in future experiments.

 (2) The authors used various sizes of females also. I understand that the statistical analysis can pick out male’s effect. However the female size may possibly make the results obscure. If they use a stable size of females, the results might be clearer.

The proposed experiment design would significantly simplify the evaluation of the experiment.Our experiment was based on an assumption that the influence of the size of the male will be more conspicuous, the greater the difference in size between the male and the female. We expected a large (positive) male effect when a large male mated with a small female and a small effect when a small male mated with a large female.

Reviewer 2 Report

This manuscript (#1916731) details a study in which the effects of female size and male size on egg production, egg mass and latency to oviposition were investigated in the firebug. Well known effects of female size were observed, but there was no effect of male size (absolute or relative to the female). The authors discuss the result in light of previous laboratory work on mating preferences.

The discussion of prior work on mating preferences needs to be clearer. According to Honek (2003), the larger of two males is more likely to mate with a large female but not a small female. In addition when 2 females and a male were used, the larger female would be the one more likely to mate, but only if the male was large. This suggests that both large males and large females have a preference (or other mating advantage) with each other, but smaller individuals have little preference, or even a preference for smaller mates.

Have the authors ruled out assortative mating by size? This perhaps could be done with a 2-factor ANOVA (male size and female size as main effects) with an interaction test. One type of significant interaction might show whether small males and females actually prefer each other over a larger individual. This provides a different test than the relative size test in the present manuscript.

There are many studies investigating the advantages of mating with an individual of larger size. The present study could do a better job of explaining why this paper is an important advance.

Table 4 had all the same information as Table 3 in the file I downloaded, so the results for latency to oviposition could not be evaluated.

Minor points:

L. 22-24. I did not think this sentence was necessary for an abstract. It is appropriate for the section on statistical analysis.

L. 40-47. I found the discussion of the benefits of accessory fluids for females unclear. The female could use these for her own benefit (for example, future survival or a later clutch) or for her eggs in her current clutch (egg number, size or quality). The current text does not seem to distinguish the benefits for the female parent from benefits for her current clutch.

Table 3. The unit for egg mass should be given. I suggest a consistent use of either egg mass or egg weight. Should it be 10^-4 instead of 10^4 in the table caption?

L. 82, 83. I suggest including the country with the place name.

How long were the males and females paired? For the duration of the experiment or was the male removed after a period of time?

Do the authors assume no reproduction in the field prior to the experiment?

What is the interpretation for the effect of female size on egg mass for only the first clutch (Table 3)? Would a larger egg size be more important early in the season?

L. 216-218. If extremely large and small males are not available in natural populations, then discussion of the effect of mate selection on those size categories is irrelevant to natural selection.

L. 220-224. Would sperm viability over so many clutches be expected to be a problem with the large local population sizes and high frequency of repeated matings? Is there information on the relationship between male size and sperm quantity in the firebug?

Author Response

Reviewer 2

Open Review

English language and style

( ) Extensive editing of English language and style required
(x) Moderate English changes required
( ) English language and style are fine/minor spell check required
( ) I don't feel qualified to judge about the English language and style

Yes

Can be improved

Must be improved

Not applicable

Does the introduction provide sufficient background and include all relevant references?

( )

(x)

( )

( )

Are all the cited references relevant to the research?

(x)

( )

( )

( )

Is the research design appropriate?

(x)

( )

( )

( )

Are the methods adequately described?

( )

(x)

( )

( )

Are the results clearly presented?

( )

(x)

( )

( )

Are the conclusions supported by the results?

( )

( )

(x)

( )

Comments and Suggestions for Authors

Minor changes to English have been corrected according to suggestions of  reviewers. The English was edited before sending to Insects, with the help of American Journal Experts. We attach a certificate of editing.

This manuscript (#1916731) details a study in which the effects of female size and male size on egg production, egg mass and latency to oviposition were investigated in the firebug. Well known effects of female size were observed, but there was no effect of male size (absolute or relative to the female). The authors discuss the result in light of previous laboratory work on mating preferences.

The discussion of prior work on mating preferences needs to be clearer. According to Honek (2003), the larger of two males is more likely to mate with a large female but not a small female. In addition when 2 females and a male were used, the larger female would be the one more likely to mate, but only if the male was large. This suggests that both large males and large females have a preference (or other mating advantage) with each other, but smaller individuals have little preference, or even a preference for smaller mates.

Many thanks for the suggestion of text improvement. The changes are in the first paragraph of the Discussion, lines 3-10.

Have the authors ruled out assortative mating by size? This perhaps could be done with a 2-factor ANOVA (male size and female size as main effects) with an interaction test. One type of significant interaction might show whether small males and females actually prefer each other over a larger individual. This provides a different test than the relative size test in the present manuscript.

Another interesting problem that unfortunately cannot be solved statistically using our results. There are no observations for some combinations of the two factor levels, male and female size. SigmaStat uses a two way design with the added assumption of no interaction between the factors.

There are many studies investigating the advantages of mating with an individual of larger size. The present study could do a better job of explaining why this paper is an important advance.

A sentence and references were added at the end of the second paragraph of rhe Introduction.

Table 4 had all the same information as Table 3 in the file I downloaded, so the results for latency to oviposition could not be evaluated.

We apologize for inserting the wrong table. Error corrected.

Minor points:

  1. 22-24. I did not think this sentence was necessary for an abstract. It is appropriate for the section on statistical analysis.

We would prefer to leave the sentence describing the statistical methods used in the study in the Abstract. This is because there are more methods available. After reading the Abstract, the reader can get an idea of which one was used.

  1. 40-47. I found the discussion of the benefits of accessory fluids for females unclear. The female could use these for her own benefit (for example, future survival or a later clutch) or for her eggs in her current clutch (egg number, size or quality). The current text does not seem to distinguish the benefits for the female parent from benefits for her current clutch.

Section rephrased

Table 3. The unit for egg mass should be given. I suggest a consistent use of either egg mass or egg weight. Should it be 10^-4 instead of 10^4 in the table caption?

Egg weight → egg mass, corrected: (g*104).

  1. 82, 83. I suggest including the country with the place name.

Done

How long were the males and females paired? For the duration of the experiment or was the male removed after a period of time?

Rephrased

Do the authors assume no reproduction in the field prior to the experiment?

Experimental animals were collected before the start of spring mating period. This information is supplemented in Material and Methods.

What is the interpretation for the effect of female size on egg mass for only the first clutch (Table 3)? Would a larger egg size be more important early in the season?

The significant relationship between the size of the female and the size of the eggs in the first clutch is very interesting and therefore we considered it important to draw attention to this relationship in the Results. However, the discussion of the significance of this interesting finding (the fact that egg size is not constrained, which is a condition for a positive relationship between female size and fecundity) is completely outside the scope of this work. Therefore, we would not like to discuss this topic in the Discussion.

  1. 216-218. If extremely large and small males are not available in natural populations, then discussion of the effect of mate selection on those size categories is irrelevant to natural selection.

The last sentence in the penultimate paragraph of the Discussion rephrased. This sentence only concerns laboratory experiments where a relationship between the size of the male and the reproduction of the female could perhaps be found in cases where males of extreme sizes would be used.

  1. 220-224. Would sperm viability over so many clutches be expected to be a problem with the large local population sizes and high frequency of repeated matings? Is there information on the relationship between male size and sperm quantity in the firebug?

A sentence added. We are not aware of any publication concerning the relationship between male size and accessory gland size. The  available information only concerns the difference in the size of the accessory glands in macropterous and brachypterous males (Socha R 2004 Decreased mating propensity of macropterous morph in a flightless wing-polymorphic insect, Pyrrhocoris apterus (Heteroptera) European Journal of Entomology 101: 539-545).

Round 2

Reviewer 2 Report

In line 206 it appears the mass of the eggs are in units of 10,000 grams.

Author Response

Egg mass, corrected  according to the opponents recommendation  (g* 10 000)
